# An Explicit Expansion of the Kullback-Leibler Divergence along its Fisher-Rao Gradient Flow

**Carles Domingo-Enrich**  *cd2754@nyu.edu*
*Courant Institute of Mathematical Sciences*
*New York University*

**Aram-Alexandre Pooladian**  *aram-alexandre.pooladian@nyu.edu*
*Center for Data Science*
*New York University*

Reviewed on OpenReview: *https://openreview.net/forum?id=9pWjgQ3y85*

## Abstract

Let $V_* : \mathbb{R}^d \to \mathbb{R}$ be some (possibly non-convex) potential function, and consider the probability measure $\pi \propto e^{-V_*}$. When $\pi$ exhibits multiple modes, it is known that sampling techniques based on Wasserstein gradient flows of the Kullback-Leibler (KL) divergence (e.g. Langevin Monte Carlo) suffer poorly in the rate of convergence, where the dynamics are unable to easily traverse between modes. In stark contrast, the work of Lu et al. (2019; 2022) has shown that the gradient flow of the KL with respect to the Fisher-Rao (FR) geometry exhibits a convergence rate to $\pi$ that is *independent* of the potential function. In this short note, we complement these existing results in the literature by providing an explicit expansion of $\mathrm{KL}(\rho_t^{\mathrm{FR}} \| \pi)$ in terms of $e^{-t}$, where $(\rho_t^{\mathrm{FR}})_{t \geq 0}$ is the FR gradient flow of the KL divergence. In turn, we are able to provide a clean asymptotic convergence rate, where the burn-in time is guaranteed to be finite. Our proof is based on observing a similarity between FR gradient flows and simulated annealing with linear scaling, and facts about cumulant generating functions. We conclude with simple synthetic experiments that demonstrate our theoretical findings are indeed tight. Based on our numerical findings, we conjecture that the asymptotic rates of convergence for Wasserstein-Fisher-Rao gradient flows are possibly related to this expansion in some cases.

## 1 Introduction

Sampling from a distribution with an unknown normalization constant is a widespread task in several scientific domains. Namely, the goal is to generate samples from a probability measure

$$\pi(x) \propto e^{-V_*(x)},$$

where $V_* : \mathbb{R}^d \to \mathbb{R}$ is some (possibly non-convex) potential function that is available for queries. In most cases, the target measure $\pi$ is only known up to the normalization constant. Applications of sampling from $\pi$ include Bayesian statistics, high-dimensional integration, differential privacy, statistical physics and uncertainty quantification; see Gelman et al. (1995); Robert et al. (1999); MacKay (2003); Johannes & Polson (2010); Von Toussaint (2011); Kobyzev et al. (2020); Chewi (2022) for thorough treatments.

Recent interest in the task of sampling stems from the following paradigm: sampling is nothing but optimization over the space of probability measures (Wibisono, 2018). This interpretation is due to the connection between the celebrated work of Jordan, Kinderleher, and Otto (Jordan et al., 1998) and the Langevin diffusion dynamics given by

$$\mathrm{d}X_t = -\nabla V_*(X_t)\,\mathrm{d}t + \sqrt{2}\,\mathrm{d}B_t, \tag{1}$$

where $dB_t$ is Brownian motion.[1] Indeed, the work of Jordan et al. (1998) demonstrates that the path in the space of proability measures given by the law of Eq. (1) is the same as the Wasserstein gradient flow (i.e. steepest descent curve in the Wasserstein metric) of the Kullback-Leibler (KL) divergence

$$\mathrm{KL}(\rho\|\pi) = \int \log \frac{\rho}{\pi} \, d\rho \,.$$

We write $(\rho_t^{\mathrm{W}})_{t\geq 0} \subseteq \mathcal{P}(\mathbb{R}^d)$ for the law of the path given by Eq. (1) (see Section 2.2.1 for a precise definition).

A central problem in this area has been to bound the convergence rate of $\rho_t^{\mathrm{W}}$ to $\pi$ in certain similarity metrics (e.g. the KL divergence itself, or the Wasserstein distance) under different conditions on $\pi$. These bounds translate to convergence rates for the Langevin Monte Carlo (LMC) sampling algorithm (Dalalyan & Tsybakov, 2012; Vempala & Wibisono, 2019; Durmus et al., 2021; Chewi et al., 2022), upon accounting for discretization errors.

The classical result is as follows: assuming that $\pi$ satisfies a Log-Sobolev inequality (LSI) with constant $C_{\mathrm{LSI}} > 0$, we obtain the following convergence rate (Stam, 1959; Gross, 1975; Markowich & Villani, 1999)

$$\mathrm{KL}(\rho_t^{\mathrm{W}}\|\pi) \leq \mathrm{KL}(\rho_0^{\mathrm{W}}\|\pi)e^{-\frac{2t}{C_{\mathrm{LSI}}}} \,, \tag{2}$$

which holds for all $t \geq 0$. Recall that $\pi$ satisfies an LSI if for all smooth test functions $g$,

$$\mathrm{ent}_\pi(f^2) \leq 2C_{\mathrm{LSI}}\mathbb{E}_\pi\|\nabla f\|^2 \,, \tag{3}$$

where $\mathrm{ent}_\pi(g) := \mathbb{E}_\pi(g \log g) - \mathbb{E}_\pi g \log \mathbb{E}_\pi g$. For example, when $V_*$ is $\alpha$-strongly convex, an LSI with $C_{\mathrm{LSI}} = 1/\alpha$ holds. LSI hold more generally, but sometimes with very large constants $C_{\mathrm{LSI}}$. Indeed, for multimodal distributions such as mixtures of Gaussians, $C_{\mathrm{LSI}}$ scales exponentially in the height of the potential barrier between modes (Holley & Stroock, 1987; Arnold et al., 2000). This impacts convergence at the discrete-time level, and thus hinders our ability to generate samples using LMC.

Another geometry that gives rise to gradient flows over probability measures is the *Fisher-Rao* (FR) geometry; see Section 2.2.2 for definitions. Similar to the case of Wasserstein gradient flows, we let $(\rho_t^{\mathrm{FR}})_{t\geq 0}$ be the FR gradient flow of the KL divergence. Recent work by Lu and collaborators has shown that the convergence $\rho_t^{\mathrm{FR}} \to \pi$ occurs at a rate that is *independent* of the potential function $V_*$. This is in stark contrast to the case of Wasserstein gradient flows, where the rate of convergence is intimately related to the structure of $V_*$ through the LSI constant. In their first work, Lu et al. (2019) show that for any $\delta \in (0, \frac{1}{4}]$ there exists a $t_* \gtrsim \log(\delta^3)$ such that for all $t \geq t_*$,

$$\mathrm{KL}(\rho_t^{\mathrm{FR}}\|\pi) \leq \mathrm{KL}(\rho_0^{\mathrm{FR}}\|\pi)e^{-(2-3\delta)(t-t_*)} \,, \tag{4}$$

where they require a warm-start condition $\mathrm{KL}(\rho_0^{\mathrm{FR}}\|\pi) \leq 1$, and assumption **(B)** (see Section 3). In Lu et al. (2022), the authors show that the KL divergence is always contracting under $(\rho_t^{\mathrm{FR}})_{t\geq 0}$ even in the absence of a warm-start, though with a worse rate. Combined, these two results provide the first continuous-time convergence rates of the gradient flow of the KL divergence under the FR geometry to $\pi$.

Merging both these geometries gives rise to the well-defined *Wasserstein-Fisher-Rao* (WFR) geometry. The WFR geometry has recently been used to analyse the convergence dynamics of parameters of neural networks (Chizat, 2022), mean-field games (Rotskoff et al., 2019), and has shown to be useful in statistical tasks such as Gaussian variational inference (Lambert et al., 2022), and identifying parameters of a Gaussian mixture model (Yan et al., 2023). In the context of sampling, particle-based methods that follow dynamics governed by WFR gradient flow of the KL, written $(\rho_t^{\mathrm{WFR}})_{t\geq 0}$, are known to escape the clutches of slow-convergence that plague the Wasserstein geometry. A simple observation (Lu et al., 2022, Remark 2.4) gives the following continuous-time convergence rate for $t \geq t_*$:

$$\mathrm{KL}(\rho_t^{\mathrm{WFR}}\|\pi) \leq \mathrm{KL}(\rho_0^{\mathrm{WFR}}\|\pi)\min\left\{e^{-C_{\mathrm{LSI}}t}, e^{-(2-3\delta)(t-t_*)}\right\} \,, \tag{5}$$

---

[1]This equation is to be understood from the perspective of Itô calculus.

where $\delta$ and $t_*$ are as in the FR convergence rate (4). Loosely speaking, this "decoupled rate" is a consequence of the Wasserstein and FR geometries being orthogonal to one another; this is made precise in Gallouët & Monsaingeon (2017).

As elegant as this last connection may seem, the convergence rate in Eq. (4), and consequently Eq. (5), should appear somewhat unsatisfactory to the reader. It raises the natural question of whether or not the factor of $\delta$ appearing in the rate is avoidable, and whether the upper bound in Eq. (4) is tight.

## 1.1 Main contributions

We close this gap for the KL divergence and any $q$-Rényi divergence. Using a different proof technique than existing work, we prove the following asymptotic rate of convergence for the flow $(\rho_t^{\mathrm{FR}})_{t \geq 0}$, namely,

$$\mathrm{KL}(\rho_t^{\mathrm{FR}} \| \pi) = \tfrac{1}{2}\mathrm{Var}_\pi \left( \log \frac{\rho_0^{\mathrm{FR}}}{\pi} \right) e^{-2t} + O(e^{-3t}) \,, \tag{6}$$

and a similar result holds for all $q$-Rényi divergences. Our assumptions are weaker to that of prior work, and given that this is a tight asymptotic convergence rate, we conjecture that the assumptions are likely unavoidable in the large $t$ regime. Our proof technique provides an explicit expansion of $\mathrm{KL}(\rho_t^{\mathrm{FR}} \| \pi)$ (and $q$-Rényi) in terms of $e^{-t}$. We supplement our finding with simulations for all three geometries, indicating that our convergence rate is in fact tight for Fisher-Rao gradient flows, and sheds light on possible conjectures for the convergence rate of WFR gradient flows.

## Notation

For a probability measure $\rho \in \mathcal{P}(\mathbb{R}^d)$ and a function $f : \mathbb{R}^d \to \mathbb{R}$, we sometimes use the shorthand $\langle f \rangle_\rho := \int f \, \mathrm{d}\rho$. We let $\log(\cdot)$ denote the natural logarithm, and we use the standard shorthand notation $f = O(g)$, meaning there exists a constant $C > 0$ such that $f \leq Cg$.

# 2 Background

## 2.1 Definitions

The study of gradient flows has a rich history in both pure and applied mathematics. The development of the relevant calculus to understand gradient flows is not the purpose of this note, and we instead provide a barebones introduction. However, we strongly recommend the interested reader consult standard textbooks on the topic, namely Ambrosio et al. (2005), and the first chapter of Chewi (2022).

Let $\mathcal{P}(\mathbb{R}^d)$ be the space of probability measures over $\mathbb{R}^d$. A functional $\mathcal{F} : \mathcal{P}(\mathbb{R}^d) \to \mathbb{R}$ is defined on the space of probability measures, with $\rho \mapsto \mathcal{F}(\rho) \in \mathbb{R}$. We call $\delta\mathcal{F}(\rho)$ the *first variation of $\mathcal{F}$ at $\rho$* if for a signed measure $\eta$ such that $\int \mathrm{d}\eta = 0$, it holds that

$$\lim_{\varepsilon \to 0} \frac{\mathcal{F}(\rho + \varepsilon\eta) - \mathcal{F}(\rho)}{\varepsilon} = \int \delta\mathcal{F}(\rho) \, \mathrm{d}\eta \,. \tag{7}$$

The Kullback-Leibler (KL) divergence of a measure $\rho$ with respect to some fixed target measure $\pi$ is defined as $\mathrm{KL}(\rho\|\pi) = \int \log \frac{\rho}{\pi} \, \mathrm{d}\rho$ for $\rho$ absolutely continuous with respect to $\pi$. For $\pi \propto e^{-V_*}$, the first variation of the KL divergence is given by

$$\delta\mathrm{KL}(\cdot\|\pi)(\rho)(x) = \log \frac{\rho(x)}{\pi(x)} = \log \rho(x) + V_*(x) + \log Z_1 \,, \tag{8}$$

where $Z_1$ is the normalizing constant for $\pi$.

A more general notion of dissimilarity between probability measures is the $q$-Rényi divergence: for $q \in [1, \infty]$, we define $\mathcal{R}_q(\rho\|\pi)$ to be the $q$-Rényi divergence with respect to $\pi$, given by

$$\mathcal{R}_q(\rho\|\pi) := \frac{1}{q-1} \log \int \left( \frac{\rho}{\pi} \right)^q \mathrm{d}\pi \,, \tag{9}$$

for measures $\rho$ that are absolutely continuous with respect to $\pi$. $\mathcal{R}_q$ recovers the KL divergence in the limit $q \to 1$, and when $q = 2$, $\mathcal{R}_2(\rho\|\pi) = \log(\chi^2(\rho\|\pi) + 1)$, where $\chi^2$ is the chi-squared divergence, written explicitly as

$$\chi^2(\rho\|\pi) = \mathrm{Var}_\pi\left(\frac{\rho}{\pi}\right) = \int \left(\frac{\rho}{\pi}\right)^2 \mathrm{d}\pi - 1\,.$$

## 2.2 Gradient flows of the Kullback-Leibler divergence

### 2.2.1 Wasserstein gradient flow

In its *dynamic formulation*, the 2-Wasserstein distance between two probability measures $\rho_0, \rho_1$ with bounded second moments can be written as (Villani, 2008; Benamou & Brenier, 2000)

$$\mathrm{W}_2^2(\rho_0, \rho_1) \coloneqq \inf_{(\rho_t, v_t)} \int_0^1 \int \|v_t(x)\|^2 \rho_t(x)\, \mathrm{d}x\, \mathrm{d}t \quad \text{s.t.} \quad \partial_t \rho_t + \nabla \cdot (\rho_t v_t) = 0\,, \tag{10}$$

where $(\rho_t)_{t \in [0,1]}$ is a curve of probability densities over $\mathbb{R}^d$, and $(v_t)_{t \in [0,1]}$ is a curve of $L^2(\mathbb{R}^d)^d$ vector fields. The constraint is known as the continuity equation, with endpoints $\rho_0$ and $\rho_1$. For a functional $\mathcal{F} : \mathcal{P}(\mathbb{R}^d) \to \mathbb{R}$, the *Wasserstein gradient flow* is the curve of measures $(\rho_t^{\mathrm{W}})_{t \geq 0}$ that satisfies the continuity equation with the vector field replaced by the steepest descent under the Wasserstein geometry,

$$v_t = -\nabla_{W_2}\mathcal{F}(\rho_t^{\mathrm{W}}) \coloneqq \nabla\delta\mathcal{F}(\rho_t^{\mathrm{W}})\,,$$

where the last equation is simply the (standard) spatial gradient of the first variation of $\mathcal{F}$. Plugging in the expression for the first variation of the KL divergence (8), we see that the law of the Langevin diffusion is given by $\rho_t^{\mathrm{W}}$ which satisfies

$$\partial_t \rho_t^{\mathrm{W}} = \nabla \cdot \left(\rho_t^{\mathrm{W}}(\nabla \log \rho_t^{\mathrm{W}} + \nabla V_*)\right)\,. \tag{11}$$

This equation may be rewritten as $\partial_t \rho_t^{\mathrm{W}} = \nabla \cdot (\nabla V_* \rho_t^{\mathrm{W}}) + \Delta \rho_t^{\mathrm{W}}$, which one readily identifies as the Fokker-Planck equation for the potential $V_*$. The equation describes the evolution of the distribution of a particle that moves according to the stochastic differential equation 1. At the particle level, the key aspect of Wasserstein gradient flows is that they model particle *transport*, and that makes them useful for high-dimensional applications such as LMC. In what follows, we will sometimes abbreviate Wasserstein gradient flow to W-GF.

### 2.2.2 Fisher-Rao gradient flow

The Fisher-Rao distance, or Hellinger-Kakutani distance, between probability measures has a long history in statistics and information theory (Hellinger, 1909; Kakutani, 1948). It can be defined as (Bogachev, 2007; Gallouët & Monsaingeon, 2017)

$$\mathrm{FR}^2(\rho_0, \rho_1) \coloneqq \inf_{(\rho_t, r_t)} \int_0^1 \int r_t(x)^2 \rho_t(x)\, \mathrm{d}x\, \mathrm{d}t \quad \text{s.t.} \quad \partial_t \rho_t = r_t \rho_t\,,$$

where $(\rho_t)_{t \in [0,1]}$ is again a curve of probability measures, and $(r_t)_{t \in [0,1]}$ is a curve of $L^2(\mathbb{R}^d)$ functions. Together, they satisfy the prescribed equation, with endpoints equal to $\rho_0$ and $\rho_1$. The Fisher-Rao gradient flow of the KL divergence, also known as *Birth-Death dynamics*, is the curve of measures $(\rho_t^{\mathrm{FR}})_{t \geq 0}$ that satisfies (Gallouët & Monsaingeon, 2017; Lu et al., 2019)

$$\partial_t \rho_t^{\mathrm{FR}} = -\rho_t^{\mathrm{FR}} \alpha_t\,, \quad \alpha_t \coloneqq \log \frac{\rho_t^{\mathrm{FR}}}{\pi} - \mathrm{KL}(\rho_t^{\mathrm{FR}}\|\pi)\,.$$

The first term adjusts mass (i.e. gives birth to or kills mass) according to the log-ratio of $\rho_t^{\mathrm{FR}}$ and the target measure $\pi$. The last term preserves the total mass, so that $\rho_t^{\mathrm{FR}} \in \mathcal{P}(\mathbb{R}^d)$ for all time.

Expanding this equation, we have

$$\partial_t \rho_t^{\mathrm{FR}}(x) = -\big( \log(\rho_t^{\mathrm{FR}}(x)) + V_*(x) - \big\langle \log(\rho_t^{\mathrm{FR}}) + V_* \big\rangle_{\rho_t^{\mathrm{FR}}} \big) \rho_t^{\mathrm{FR}}(x). \tag{12}$$

We henceforth omit the superscript FR for the Fisher-Rao gradient flow of the KL divergence unless the notation becomes ambiguous. For short-hand, we make use of the abbreviation FR-GF for Fisher-Rao gradient flows.

The FR-GF may be simulated using a system of weighted particles (see Appendix B). Unlike for the W-GF, in this case the positions of the particles are fixed; only the weights change over time. Hence, to simulate the FR-GF one is forced to grid the underlying space $\mathbb{R}^d$. This is feasible only for small dimensions $d$. Consequently, FR-GFs cannot be simulated in high dimensions, which makes them impractical for sampling applications.

### 2.2.3 Wasserstein-Fisher-Rao geometry gradient flow

The Wasserstein-Fisher-Rao distance between probability measures arises as a combination of the Wasserstein and the Fisher-Rao distances (Chizat et al., 2018; 2015; Kondratyev et al., 2016; Liero et al., 2016; 2018). It is defined as

$$\mathrm{WFR}^2(\rho_1, \rho_2) := \inf_{(\rho_t, v_t, r_t)} \int_0^1 \int (\|v_t(x)\|^2 + r_t(x)^2) \rho_t(x) \, \mathrm{d}x \, \mathrm{d}t \quad \text{s.t.} \quad \partial_t \rho_t + \nabla \cdot (\rho_t v_t) = r_t \rho_t \,,$$

where, for each $t \in [0,1]$, the triple $(\rho_t, v_t, r_t)$ lives in $\mathcal{P}(\mathbb{R}^d) \times L^2(\mathbb{R}^d)^d \times L^2(\mathbb{R}^d)$, and they simultaneously satisfy the constraint equation, which has endpoints $\rho_0$ and $\rho_1$, as well. Similarly, the Wasserstein-Fisher-Rao gradient flow of the KL divergence is the solution of PDE that incorporates the terms in the Wasserstein and Fisher-Rao gradient flows (Eq. (11) and Eq. (12)):

$$\partial_t \rho_t^{\mathrm{WFR}} = \nabla \cdot \big( \rho_t^{\mathrm{WFR}}(\nabla \log \rho_t^{\mathrm{WFR}} + \nabla V_*) \big) - \big( \log(\rho_t^{\mathrm{WFR}}) + V_* - \big\langle \log(\rho_t^{\mathrm{WFR}}) + V_* \big\rangle_{\rho_t^{\mathrm{WFR}}} \big) \rho_t^{\mathrm{WFR}} \tag{13}$$

Similar to the other geometries, we write WFR-GF as shorthand for Wasserstein-Fisher-Rao gradient flow At the particle level, WFR-GFs are able to capture both *transport* and *weight updates*, which is why they enjoy a convergence rate that at least matches the better rate between W- and FR-GFs (recall Eq. (5)), and is clearly superior in practice in some instances. Hence, any improvement in the convergence analysis of either W- or FR-GFs translates to improving our understanding of WFR-GFs.

### 2.3 Simulated annealing dynamics

Simulated annealing is a technique seen in several works when attempting to either optimize a function or sample from a multimodal probability distribution, and has a long history (Pincus, 1970; Kirkpatrick et al., 1983), and plays a crucial role in our analysis. In what follows, we introduce the annealing path with linear scaling, and conclude with a proposition.

Consider the time-dependent measure $(\mu_\tau)_{\tau \in [0,1]}$ corresponding to the annealing path, with *linear scaling*, initialized at the measure $\mu_0 = \rho_0 \propto e^{-V_0}$. By definition, $\mu_\tau$ admits the density

$$\mu_\tau(x) = \frac{e^{-\tau(V_*(x) - V_0(x)) - V_0(x)}}{Z_\tau}, \quad Z_\tau = \int_{\mathbb{R}^d} e^{-\tau(V_*(x) - V_0(x)) - V_0(x)} \, dx, \tag{14}$$

for $\tau \in [0,1]$. Note that indeed, $\mu_1 = \pi$. To this end, it will be convenient to rewrite Eq. (14) in terms of the log-density of $\mu_\tau$. Remark that

$$\log(\mu_\tau(x)) = -\tau(V_*(x) - V_0(x)) - V_0(x) - \log Z_\tau \,. \tag{15}$$

One can check that the pointwise derivative of the density $\mu_\tau$ (with respect to $\tau$) is

$$\partial_\tau \mu_\tau(x) = -(V_*(x) - V_0(x) - \langle V_* - V_0 \rangle_{\mu_\tau}) \mu_\tau(x) \,. \tag{16}$$

From this, we obtain that

$$
\begin{aligned}
&\log(\mu_\tau(x)) + V_*(x) - \big\langle \log(\mu_\tau) + V_* \big\rangle_{\mu_\tau} \\
&= -\tau(V_*(x) - V_0(x)) - V_0(x) - \big\langle -\tau(V_* - V_0) - V_0 + V_* \big\rangle_{\mu_\tau} + V_*(x) - \langle V_* \rangle_{\mu_\tau} \\
&= -\tau(V_*(x) - V_0(x)) - V_0(x) + V_*(x) - \big\langle -\tau(V_* - V_0) - V_0 + V_* \big\rangle_{\mu_\tau} \\
&= (1-\tau)\big(V_*(x) - V_0(x)\big) - (1-\tau)\langle V_* - V_0 \rangle_{\mu_\tau} \\
&= (1-\tau)\big(V_*(x) - V_0(x) - \langle V_* - V_0 \rangle_{\mu_\tau}\big).
\end{aligned}
\tag{17}
$$

Note that in the first equality, we used that the log-partition is a constant and gets cancelled out by the difference of the two terms. Consequently, Eq. (16) can be rewritten, for $\tau \in (0,1)$, as

$$
\partial_\tau \mu_\tau(x) = -\frac{1}{1-\tau}\big(\log(\mu_\tau(x)) + V_*(x) - \big\langle \log(\mu_\tau) + V_* \big\rangle_{\mu_\tau}\big)\mu_\tau(x).
\tag{18}
$$

A first observation is that that the linear schedule $\tau$ in the exponent of Eq. (14) results in dynamics that resemble the Fisher-Rao gradient flow of the KL divergence, up to a reparameterization that can be made explicit. Indeed, if one compares Eq. (18) with Eq. (12), the only difference is the factor $\frac{1}{1-\tau}$ in the right-hand side of Eq. (18). Since the solution of the Fisher-Rao gradient flow of the KL divergence is unique (see Proposition 4 in Appendix A), an appropriate time reparameterization of the annealed dynamics (14) will yield the solution (12). We summarize this observation in the following proposition, which we were unable to find a citation for in the literature.

**Proposition 1.** *Let $(\mu_\tau)_{\tau\in[0,1]}$ be as defined in Eq. (14). The Fisher-Rao gradient flow $(\rho_t)_{t\geq 0}$ of $KL(\rho\|\pi)$ (i.e. solving Eq. (12)) is given by $\rho_t = \mu_{1-e^{-t}}$.*

*Proof.* If we write $t$ as a function of $\tau$, we have that

$$
\partial_\tau \rho_{t(\tau)} = \partial_t \rho_{t(\tau)} \frac{dt}{d\tau}(\tau) = -\frac{dt}{d\tau}(\tau)\big(\log(\rho_{t(\tau)}(x)) + E(x) - \big\langle \log(\rho_{t(\tau)}) + E \big\rangle_{\rho_{t(\tau)}}\big)\rho_{t(\tau)}(x).
\tag{19}
$$

Identifying $\rho_{t(\tau)}$ with $\rho_\tau$, and establishing a direct comparison with Eq. (18), we obtain that for Eq. (19) to hold, $t(\tau)$ must fulfill $\frac{dt}{d\tau}(\tau) = \frac{1}{1-\tau}$. With the initial condition that $\tau(0) = 0$, this differential equation has the following unique solution:

$$
t(\tau) = \int_0^\tau \frac{1}{1-s}\, ds = -\log(1-\tau).
\tag{20}
$$

That is, we have that $t(\tau) = -\log(1-\tau)$, or equivalently, $\tau(t) = 1 - e^{-t}$. $\qquad\square$

## 2.4 Cumulants and their power series

Our core argument hinges on observing a relation between the above gradient flows and their connection to cumulants of a random variable. Recall that for a random variable $Y$, its *cumulant-generating function* to be $K_Y(z) = \log \mathbb{E}[e^{Yz}]$. The $n^{\text{th}}$ cumulant $\kappa_n$ of the random variable $Y$ is defined as the $n^{\text{th}}$ derivative of $K_Y$ evaluated at $z = 0$, that is, $\kappa_n = K_Y^{(n)}(0)$. Similar to moment-generating functions, if $K_Y(z)$ is finite in some neighborhood of $z \in (-\epsilon_0, \epsilon_0)$, then it holds that $K_Y$ is smooth (in fact, holomorphic) (see e.g. (Shiryaev, 1984, Section II.12.8). Moreover, $K_Y(z)$ admits the following infinite series expansion

$$
K_Y(z) = \sum_{n\geq 1} \frac{\kappa_n}{n!} z^n.
$$

In particular, one can easily check that $\kappa_1 = \mathbb{E}[Y]$ and $\kappa_2 = \mathrm{Var}(Y)$.

## 3 Main result

The goal of this section is to prove our main result, which is an explicit expansion of the KL divergence in terms of log-cumulants of the random variable $\log \frac{\rho_0(X)}{\pi(X)}$ where $X \sim \pi$. We make the following assumptions throughout, and we will make their uses explicit when necessary.

**(A1)** $V_* \in L_1(\pi)$,

**(A2)** There exists $\alpha \in \mathbb{R}_+$, such that $\inf_x \frac{\rho_0(x)}{\pi(x)^{1+\alpha}} > 0$.

Assumption **(A1)** ensures that $\pi$ has finite differential entropy, and is a relatively weak condition. **(A2)** asks that at least some mass is initially placed along the support of $\pi$. **(A2)** is, however, a much weaker assumption that what is currently used in the literature. To be precise, Lu et al. (2019; 2022) assume a particular case of **(A2)**, namely

**(B)** There exists $M > 0$ such that $\inf_x \frac{\rho_0(x)}{\pi(x)} \geq e^{-M}$.

This is the same as **(A2)** when $\alpha$ is constrained to be 0, and they make explicit use of the constant $M > 0$ in their rate. Note that **(A2)** is weaker the larger $\alpha$ is, as $\pi(x)^{1+\alpha}$ decreases faster. As a comparison, if $\rho_0$ and $\pi$ are Gaussians, **(A2)** covers the setting where both have arbitrary means and covariances, while constraining $\alpha = 0$ only covers the cases in which the covariance matrix of $\rho_0$ is strictly larger than the one of $\pi$ in the positive definite order.

The following theorem is our main contribution. While here we have stated an asymptotic expression, in fact a more general expression is available as an infinite power series for times large enough, and appears explicitly in the proof (Eq. (29) and Eq. (30)).

**Theorem 1.** *Suppose **(A1)** and **(A2)** hold. Then for any $q \in (1, \infty)$,*

$$KL(\rho_t \| \pi) = \frac{\kappa_2}{2} e^{-2t} + O(e^{-3t}), \quad \text{and} \quad \mathcal{R}_q(\rho_t \| \pi) = \frac{q\kappa_2}{2} e^{-2t} + O_q(e^{-3t}), \tag{21}$$

*where $\kappa_2 = Var_\pi \left( \log \frac{\rho_0}{\pi} \right)$. The remainder terms $O(e^{-3t})$ and $O_q(e^{-3t})$ depend on the initialization $\rho_0$ and on the target $\pi$.*

Note that the result (4) by Lu et al. (2019) implies an asymptotic rate very close to $e^{-2t}$, but there are significant differences between both: beyond the fact that our result holds under much weaker assumptions, we characterize exactly the asymptotic decay of $KL(\rho_t \| \pi)$, while they only provide an upper-bound that becomes less tight as $\delta$ goes to zero (because the constant $t_*$ increases).

**Remark 1.** *The coefficient $\kappa_2$ is nothing more than the variance under $\pi$ of the first-variation of the KL divergence at $\rho_0$ (recall Eq. (8)).*

### 3.1 Proof

Given potentials $V_*$ and $V_0$ such that $\pi \propto e^{-V_*}$ and $\rho_0 \propto e^{-V_0}$, we define the random variable

$$Y := V_*(X) - V_0(X) \text{ where } X \sim \pi, \tag{22}$$

Note that we can set $V_*(x) = -\log \pi(x)$ and $V_0(x) = -\log \rho_0(x)$, which means that $Y = \log \frac{\rho_0(X)}{\pi(X)}$, but adding any constant term to $V_*$ and $V_0$ (or solely $V_0$) also yields a valid construction of $Y$.

**Proposition 2.** *Let $Y$ be as in Eq. (22). Let $(\mu_\tau)_{\tau \in [0,1]}$ be follow the simulated annealing dynamics from Eq. (14). It holds that*

$$KL(\mu_\tau \| \pi) = (1 - \tau) K_Y'(1 - \tau) - K_Y(1 - \tau), \tag{23}$$

$$\mathcal{R}_q(\mu_\tau \| \pi) = \frac{1}{q-1} K_Y(q(1 - \tau)) - \frac{q}{q-1} K_Y(1 - \tau). \tag{24}$$

*Proof.* We first identify the following relationship, which arises from a simple manipulation of Eq. (14)

$$K_Y(1-\tau) = \log\left(\int e^{(1-\tau)(V_*(x)-V_0(x))}\frac{e^{-V_*(x)}}{Z_1}\,dx\right) = \log\left(\frac{\int e^{-\tau(V_*(x)-V_0(x))-V_0(x)}\,dx}{Z_1}\right) = \log Z_\tau - \log Z_1.$$
(25)

Using this expression, we can expand the KL divergence between $\mu_\tau$ and $\pi$ as follows:

$$\mathrm{KL}(\mu_\tau\|\pi) = \int \log\frac{\mu_\tau}{\pi}\mu_\tau = \int \log\left(\frac{e^{-\tau(V_*-V_0)-V_0}Z_\tau^{-1}}{e^{-V_*}Z_1^{-1}}\right)\,\mathrm{d}\mu_\tau$$
$$= \log Z_1 - \log Z_\tau + (1-\tau)\langle V_* - V_0\rangle_{\mu_\tau}$$
$$= (1-\tau)\langle V_* - V_0\rangle_{\mu_\tau} - K_Y(1-\tau)\,.$$

Another fact about cumulant generating functions that we can exploit is the following differential relationship

$$-\langle V_* - V_0\rangle_{\mu_\tau} = \frac{\mathrm{d}}{\mathrm{d}\tau}Z_\tau = -K_Y'(1-\tau)\,.$$
(26)

Altogether, this gives

$$\mathrm{KL}(\mu_\tau\|\pi) = (1-\tau)K_Y'(1-\tau) - K_Y(1-\tau)\,.$$
(27)

The general $q$-Rényi case is deferred to the appendix, where the computation is similar. $\square$

The following lemma uses both **(A1)** and **(A2)** to establish that $K_Y(z)$ is finite in some neighborhood of $z \in B_{\epsilon_0}(0)$, which implies that $K_Y$ admits the series expansion we will require in the sequel. The proof is deferred to the appendix.

**Proposition 3.** *Suppose **(A1)** and **(A2)** are satisfied. Then there exists some constant $\epsilon_0 > 0$ such that the cumulant generating function of $Y$, $K_Y(z) = \log\mathbb{E}[e^{Yz}]$ is finite on some neighborhood of $z \in B_{\epsilon_0}(0)$. Moreover, inside this neighborhood, $K_Y(z)$ is holomorphic and we have the series expansion*

$$K_Y(z) = \sum_{n\geq 1}\frac{\kappa_n}{n!}z^n\,.$$
(28)

We conclude with the proof of our main result.

*Proof of Theorem 1.* We begin with the expression of the KL divergence. Note that since $K_Y(z)$ is smooth for $z$ sufficiently close to the origin, it holds that

$$K_Y'(z) = \sum_{n\geq 1}\frac{\kappa_n}{(n-1)!}z^{n-1}\,.$$

Using the parameterization of Eq. (27) and the series expansion for $K_Y'(1-\tau)$, our expression for $\mathrm{KL}(\mu_\tau\|\pi)$ reads

$$\mathrm{KL}(\mu_\tau\|\pi) = (1-\tau)\sum_{n\geq 1}\frac{\kappa_n}{(n-1)!}(1-\tau)^{n-1} - \sum_{n\geq 1}\frac{\kappa_n}{n!}(1-\tau)^n$$
$$= \sum_{n\geq 1}\kappa_n\left(\frac{n}{n!} - \frac{1}{n!}\right)(1-\tau)^n$$
$$= \sum_{n\geq 2}\frac{\kappa_n}{n(n-2)!}(1-\tau)^n\,.$$

Expanding the relation and replacing $\tau(t) = 1 - e^{-t}$ gives

$$\mathrm{KL}(\rho_t\|\pi) = \frac{\kappa_2}{2}e^{-2t} + \sum_{n\geq 3}\frac{\kappa_n}{n(n-2)!}e^{-nt}\,.$$
(29)

We now do the same manipulations for $\mathcal{R}_q(\mu_\tau\|\pi)$.

$$\mathcal{R}_q(\mu_\tau\|\pi) = \frac{1}{q-1}\sum_{n\geq 1}\frac{\kappa_n}{n!}(q(1-\tau))^n - \frac{q}{q-1}\sum_{n\geq 1}\frac{\kappa_n}{n!}(1-\tau)^n$$

$$= \frac{1}{q-1}\left(\frac{\kappa_1}{q}(1-\tau) + \sum_{n\geq 2}q^n\frac{\kappa_n}{n!}(1-\tau)^n\right) - \frac{q}{q-1}\left(\kappa_1(1-\tau) + \sum_{n\geq 2}\frac{\kappa_n}{n!}(1-\tau)^n\right)$$

$$= \sum_{n\geq 2}\frac{q^n - q}{q-1}\frac{\kappa_n}{n!}(1-\tau)^n.$$

Substituting $\tau(t) = 1 - e^{-t}$ and expanding out the first term yields

$$\mathcal{R}_q(\rho_t\|\pi) = q\frac{\kappa_2}{2}e^{-2t} + \sum_{n\geq 3}\frac{q^n - q}{q-1}\frac{\kappa_n}{n!}e^{-nt}. \tag{30}$$

Lemma 2 in the appendix justifies that the higher-order terms of Eq. (29) and Eq. (30) are $O(e^{-3t})$, which concludes the proof. $\qquad\square$

## 4 Numerical simulations

We present simple numerical simulations that demonstrates our asymptotic convergence rate of the KL divergence the FR gradient flows, as well as a comparison with the WFR- and W-GFs. We consider two target distributions over the set $[-\pi, \pi)$, each with two initializations:

1. Target distribution $\pi_1$: We set $\pi_1 \propto e^{-V_1}$ with $V_1(x) = 2.5\cos(2x) + 0.5\sin(x)$. This distribution has two modes with different weights and has been studied previously by Lu et al. (2019). We consider two initial distributions:

   (a) $\pi_a \propto e^{-V_a}$ with $V_a = -V_1$, which has two modes in locations where $\pi$ has little mass.
   (b) $\pi_b \propto e^{-V_b}$ with $V_b = 2.5\cos(2x)$, which has two modes in almost the same positions as $\pi$, but with equal weight.

2. Target distribution $\pi_2$: We set $\pi_2 \propto e^{-V_2}$ with $V_2(x) = -6\cos(x)$. This distribution has one mode. We consider two initial distributions:

   (c) $\pi_c \propto e^{-V_c}$ with $V_c = -V_2$, which has one mode in a location where $\pi$ has little mass.
   (d) $\pi_d \propto e^{-V_d}$ with $V_d = 0$, which is the uniform distribution.

Fig. 1 shows the target energies $V_1$, $V_2$ and the initial energies $V_a$, $V_b$, $V_c$, $V_d$ introduced above. Fig. 2 shows the evolution of the KL divergence along the FR, WFR and W gradient flows. It also contains plots of the dominant term $\frac{\kappa_2}{2}e^{-2t}$ of the approximation of the KL divergence decay for FR flows (see Theorem 1), displayed as dotted lines. Table 1 shows the slopes of each curve from Fig. 2, at large times (see Appendix B for details on the computation of slopes).

Some observations are in order:

- As predicted by Theorem 1, the curves $\mathrm{KL}(\rho_t^{\mathrm{FR}}\|\pi)$ approach the curves $\frac{\kappa_2}{2}e^{-2t}$ as $t$ grows.

- For $\pi_1$, the curves $\mathrm{KL}(\rho_t^{\mathrm{FR}}\|\pi)$ and $\mathrm{KL}(\rho_t^{\mathrm{WFR}}\|\pi)$ initialized at $\pi_b$ are very close for small times. The reason is that $\nabla V_1$ and $\nabla V_b$ are very close in the regions where $\pi_1$ and $\pi_b$ have most of the mass. Consequently, the term $\nabla\cdot\left(\rho_t^{\mathrm{WFR}}(\nabla\log\rho_t^{\mathrm{WFR}} + \nabla V_1)\right)$, which is the difference between the FR and the WFR PDEs, is small at initialization.

- The curves $\mathrm{KL}(\rho_t^{\mathrm{W}}\|\pi)$ behave very differently for $\pi_1$ and $\pi_2$ (see Table 1). Indeed, since $\pi_1$ is bimodal $C_{\mathrm{LSI}}(\pi_1)$ is quite large (thus convergence is slow), whereas $\pi_2$ is unimodal, with a much smaller log-Sobolev constant.

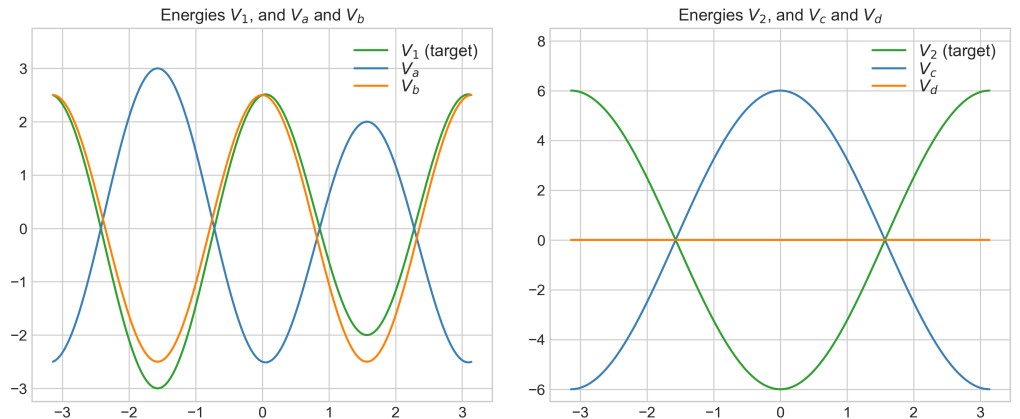

Figure 1: Energies of the target and initial distributions.

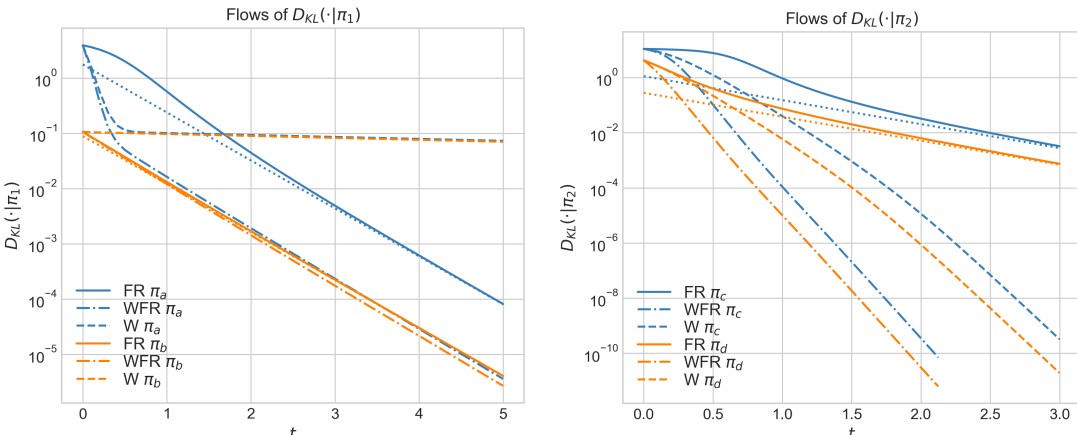

Figure 2: Evolution of the KL divergence with respect to $\pi_1$ (*left*) and $\pi_2$ (*right*) along their respective FR (*solid lines*), WFR (*dash-dotted lines*) and W (*dashed lines*) gradient flows. Each plot contains flows initialized at two probability measures: in the left plot these are $\pi_a$ (*blue*, top curves at $t = 0$) and $\pi_b$ (*orange*); in the right plot, $\pi_c$ (*blue*, top curves at $t = 0$) and $\pi_d$ (*orange*). The *dotted* lines show the curves $\frac{\kappa_2}{2} e^{-2t}$ (for the appropriate values $\kappa_2$), introduced in Theorem 1.

- The curves $\mathrm{KL}(\rho_t^{\mathrm{WFR}}\|\pi)$ also behave differently for both target distributions. For $\pi_1$, it decays only slightly faster than $\mathrm{KL}(\rho_t^{\mathrm{FR}}\|\pi)$, while for $\pi_2$ it goes down much faster than both $\mathrm{KL}(\rho_t^{\mathrm{FR}}\|\pi)$ and $\mathrm{KL}(\rho_t^{\mathrm{WFR}}\|\pi)$. Interestingly, looking at Table 1 we observe that the asymptotic slopes of the WFR are very close to the sum of the slopes for FR and W. This seems to indicate that at large times, the KL divergence decays like $e^{-2t-\frac{2t}{C_{\mathrm{LSI}}}}$, i.e. that the W and FR terms act more or less independently.

| | Target $\pi_1$ | | Target $\pi_2$ | |
|---|---|---|---|---|
| | Init. $\pi_a$ | Init. $\pi_b$ | Init. $\pi_c$ | Init. $\pi_d$ |
| FR | -2.0016 | -2.0002 | -2.0028 | -2.0014 |
| WFR | -2.0771 | -2.0759 | -12.8190 | -12.8632 |
| W | -0.0811 | -0.0811 | -10.7784 | -10.8538 |

Table 1: Large-time slopes of the KL divergence vs. time curves in a semi-logarithmic plot (Fig. 2), for the three flows. See Appendix B for details on the computation of the slopes.

## 5    Conclusion

In this work, using a relatively simple proof technique, we showed that the Kullback-Leibler divergence along its Fisher-Rao gradient flow $(\rho_t^{\mathrm{FR}})_{t \geq 0}$ can be written as a power-series expansion, resulting in a tight asymptotic convergence rate for large times. A similar expansion holds for $\mathcal{R}_q(\rho_t^{\mathrm{FR}} \| \pi)$, where $\mathcal{R}_q$ is any $q$-Rényi divergence. Our findings were verified with simple numerical experiments, where we also simulated Wasserstein and Wasserstein-Fisher-Rao gradient flows. Our simulations indicated that, in some cases, the convergence rate of the WFR gradient flow scales like $e^{-(2+(2/C_{\mathrm{LSI}}))t}$, an observation that we hope can be made precise in future work. A second direction is to extend our proof technique from the KL divergence to general Bregman divergences.

### Acknowledgments

AAP thanks Anna Korba for introducing the main WFR references to him. The authors collectively thank Joan Bruna, Jonathan Niles-Weed, Sinho Chewi, and Andre Wibisono for helpful discussions, and the anonymous reviewers who helped improve the quality of the presentation. CD acknowledges Meta AI Research as a funding source. AAP acknowledges NSF Award 1922658 and Meta AI Research.

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

## A    Remaining proofs

**Proposition 4** (Uniqueness of the Fisher-Rao gradient flow of the KL divergence). *Given a target potential $V_*$ and an initial measure $\rho_0$, the solution of Eq. (12) is unique.*

*Proof.* Consider the PDE

$$\partial_t \mu_t(x) = -\big(\log(\mu_t(x)) + V_*(x)\big)\mu_t(x), \qquad \mu_0 = \rho_0 \tag{31}$$

Note that this is in fact an ODE for each point $x$, that we can rewrite as $\partial_t \log \mu_t(x) = -\big(\log(\mu_t(x)) + V_*(x)\big)$. The unique solution of this ODE with initial condition $\log \mu_0(x)$ is $\log \mu_t(x) = (\log \mu_0(x) - V_*(x))e^{-t} + V_*(x)$. Thus, we conclude that Eq. (31) has a unique solution.

Now, given a solution $\rho_t$ of Eq. (12) with initial condition $\rho_0$, define $\tilde{\rho}_t$ as

$$\log \tilde{\rho}_t(x) = \log \rho_t(x) + \int_0^t e^{t-s} \big\langle \log(\rho_s) + V_* \big\rangle_{\rho_s} ds \tag{32}$$

Note that $R_t := -\int_0^t e^{t-s}\big\langle \log(\rho_s) + V_*\big\rangle_{\rho_s} ds$ satisfies the ODE $\frac{dR_t}{dt} = -R_t - \big\langle \log(\rho_t) + V_*\big\rangle_{\rho_t}$ with the initial condition $R_0 = 0$. Using this, we observe that $\tilde{\rho}_t$ is a solution of Eq. (31):

$$\partial_t \log \tilde{\rho}_t(x) = \partial_t \log \rho_t(x) + \partial_t R_t = -\big(\log(\rho_t(x)) + V_*(x) - \big\langle \log(\rho_t) + V_*\big\rangle_{\rho_t}\big) + \partial_t R_t$$
$$= -\big(\log(\rho_t(x)) + R_t + V_*(x)\big) = -\big(\log(\tilde{\rho}_t(x)) + V_*(x)\big).$$

Also, note that the map $(\rho_t)_{t\geq 0} \to (\tilde{\rho}_t)_{t\geq 0}$ defined by Eq. (32) is invertible, as $\rho_t(x) = \tilde{\rho}_t(x)/\int \tilde{\rho}_t(y)\,dy$. This follows from the fact that $\rho_t$ and $\tilde{\rho}_t$ are proportional to each other, and that $\rho_t$ integrates to 1.

Finally, suppose that $\rho_t^a$ and $\rho_t^b$ are two solutions of Eq. (12) with initial condition $\rho_0$. Via the construction Eq. (32), they yield solutions $\tilde{\rho}_t^a$ and $\tilde{\rho}_t^b$ of Eq. (31) with initial condition $\rho_0$. The uniqueness of the solution of Eq. (31) implies that $\tilde{\rho}_t^a = \tilde{\rho}_t^b$. Since the map $(\rho_t)_{t\geq 0} \to (\tilde{\rho}_t)_{t\geq 0}$ is invertible, we obtain that $\rho_t^a = \rho_t^b$, which concludes the proof. $\square$

*Proof of Proposition 2 (Continued).* We perform similar manipulations as in the case with the KL divergence:

$$
\begin{aligned}
\mathcal{R}_q(\mu_\tau \| \pi) &= \frac{1}{q-1} \log \int \frac{e^{-q\tau(V_* - V_0) - qV_0}(Z_\tau)^{-q}}{e^{-qV_*}(Z_1)^{-q}} \, d\pi \\
&= \frac{1}{q-1} \log \int e^{q(1-\tau)(V_* - V_0)} \left( \frac{Z_\tau}{Z_1} \right)^q d\pi \\
&= \frac{1}{q-1} K_Y(1-\tau) - \frac{q}{q-1}(\log Z_\tau - \log Z_1) \\
&= \frac{1}{q-1} K_Y(1-\tau) - \frac{q}{q-1} K_Y(1-\tau),
\end{aligned}
$$

where in the last line we again used Eq. (25). This completes the proof. $\qquad\square$

*Proof of Proposition 3.* By **(A1)**, the partition function $F(t) = \int_{\mathbb{R}^d} e^{-tV_*(x)} \, dx$ is differentiable at $t = 1$. This is because $F'(t) = -\int V_*(x) \, d\pi(x)$. Hence, $F(t)$ is finite on an interval $(1 - 2\epsilon_1, 1]$ for some $\epsilon_1$.

Note that the assumption **(A2)** can be written equivalently as $\xi := \inf_x \alpha V_*(x) - V_0(x) > -\infty$. We obtain that for all $\epsilon \in [0, \epsilon_1/\alpha)$,

$$
\begin{aligned}
-\epsilon(V_*(x) - V_0(x)) - V_*(x) &= -\epsilon((1+\alpha)V_*(x) - V_0(x)) + (\epsilon\alpha - 1)V_*(x) \\
&\leq -\epsilon\xi + (\epsilon\alpha - 1)V_*(x) \leq -\epsilon\xi + (\epsilon_1 - 1)V_*(x)
\end{aligned}
\tag{33}
$$

Equivalently,

$$
\exp(K_Y(-\epsilon)) = \int_{\mathbb{R}^d} e^{-\epsilon(V_*(x) - V_0(x)) - V_*(x)} \, dx \leq e^{-\epsilon\xi} \int_{\mathbb{R}^d} e^{-(1-\epsilon_1)V_*(x)} \, dx = e^{-\epsilon\xi} F(1 - \epsilon_1) < +\infty.
\tag{34}
$$

Also, for all $\epsilon \in [0, 1)$, using the convexity of the exponential function we have that

$$
\exp(K_Y(\epsilon)) = \int_{\mathbb{R}^d} e^{\epsilon(V_*(x) - V_0(x)) - V_*(x)} \, dx = \int_{\mathbb{R}^d} e^{-(1-\epsilon)V_*(x) - \epsilon V_0(x)} \, dx
\tag{35}
$$

$$
\leq \int_{\mathbb{R}^d} (1 - \epsilon) e^{-V_*(x)} + \epsilon e^{-V_0(x)} \, dx = (1 - \epsilon)Z_1 + \epsilon Z_0 < +\infty.
\tag{36}
$$

Hence, the cumulant-generating function $K_Y(t) = \log \mathbb{E} e^{tY}$ is finite on a neighborhood $(-\epsilon_0, \epsilon_0)$ with $\epsilon_0 = \min\{1, \epsilon_1/\alpha\}$. Applying Lemma 1, we conclude that there exists $\epsilon > 0$ such that for $z \in B_\epsilon(0)$, we have that $K_Y(z) = \sum_{n=1}^{+\infty} \frac{\kappa_n}{n!} z^n$. $\qquad\square$

The following lemma, which we make explicit, is a well-known fact in probability theory. In short, since the moment-generating function is analytic in some neighborhood, and is non-negative, taking the logarithm is safe as everything is analytic. The interested reader can consult e.g. (Shiryaev, 1984, Section II.12.8) which dissects this in detail.

**Lemma 1.** *Assume that the cumulant-generating function $K_Y(t) = \log \mathbb{E} e^{tY}$ is finite on a neighborhood $(-\epsilon_0, \epsilon_0)$ of zero. Then, $K_Y(z) = \log \mathbb{E} e^{zY}$ as a function on the complex plane is holomorphic on the open ball $B_\epsilon(0)$ of radius $\epsilon$ centered at zero, for some $\epsilon > 0$. Moreover, for $z \in B_\epsilon(0)$, we have that*

$$
K_Y(z) = \sum_{n=1}^{+\infty} \frac{\kappa_n}{n!} z^n.
\tag{37}
$$

**Lemma 2** (End of the proof of Theorem 1). *We have that*

$$
\left| KL(\rho_t \| \pi) - \frac{\kappa_2}{2} e^{-2t} \right| = O(e^{-3t}), \qquad \left| \mathcal{R}_q(\rho_t \| \pi) - \frac{q\kappa_2}{2} e^{-2t} \right| = O(e^{-3t}).
\tag{38}
$$

*Proof.* Lemma 1 implies that the series for $K_Y$ centered at zero has convergence radius $\epsilon$, for some $\epsilon > 0$. Since the derivative of a series has the same radius of convergence, we obtain that

$$H(z) := zK_Y'(z) - K_Y(z) = \sum_{n \geq 2} \frac{\kappa_n}{n(n-2)!} z^n.$$

has convergence radius $\epsilon$ as well. Hence, by the Cauchy-Hadamard theorem, $\frac{1}{\epsilon} \geq \limsup_{n \to \infty}(|c_n|^{1/n})$, where $c_n := \frac{\kappa_n}{n(n-2)!}$.

This implies that for all $0 < \epsilon' < \epsilon$, there exists a constant $C_{\epsilon'} > 0$ such that for all $n \geq 0$, $|c_n| \leq C_{\epsilon'}/(\epsilon')^n$. Consequently, for all $z \in \mathbb{C}$ with $|z| < 1/\epsilon'$,

$$|H(z) - \frac{\kappa_2}{2} z^2| = \Big| \sum_{n=3}^{+\infty} \frac{\kappa_n}{n(n-2)!} z^n \Big| \leq C_{\epsilon'} \sum_{n=3}^{+\infty} \left(\frac{|z|}{\epsilon'}\right)^n = C_{\epsilon'} \frac{\left(\frac{|z|}{\epsilon'}\right)^3}{1 - \frac{|z|}{\epsilon'}} \tag{39}$$

Using Eq. (23), we get that for any constant $\gamma > 0$, if $t \geq -\log \epsilon' + \gamma$ (or equivalently, $e^{-t} \leq \epsilon' e^{-\gamma}$),

$$|\mathrm{KL}(\rho_t \| \pi) - \frac{\kappa_2}{2} e^{-2t}| \leq C_{\epsilon'} \frac{\left(\frac{e^{-t}}{\epsilon'}\right)^3}{1 - \frac{e^{-t}}{\epsilon'}} = C_{\epsilon'} \frac{e^{-3t}}{(\epsilon')^3(1 - e^{-\gamma})} = O(e^{-3t}), \tag{40}$$

which concludes the proof for the KL divergence. For the Rényi divergence, the proof is analogous (note that in that case the series $\frac{1}{q-1} K_Y(qz) - \frac{q}{q-1} K_Y(z)$ has convergence radius $\epsilon/q$). $\square$

## B  Details on the numerical simulations

To run the simulations in Section 4, we discretized the interval $[-\pi, \pi)$ in $n = 2000$ equispaced points. Let $h = 2\pi/n$. For each algorithm and initialization, we construct sequences $(x_k)_{k \geq 0}$, where $x_k \in \mathbb{R}^n$ represents the normalized log-density at each point. We let $v_* \in \mathbb{R}^n$ be the (non-normalized) energy of the target distribution, obtained by evaluating $V_*$ at the discretization points. Similarly, $\nabla v_*, \Delta v_* \in \mathbb{R}^n$ are the evaluations of $\nabla V_*$ and $\Delta V_*$ at the $n$ points (note that $\nabla V_*$ is a scalar because the distributions are one-dimensional).

We used the following discretizations for the Fisher-Rao, Wasserstein and Wasserstein-Fisher-Rao gradient flows:

(i) Fisher-Rao GF: We use mirror descent in log-space. The update reads:

$$\tilde{x}_{k+1} \leftarrow x_k + \epsilon(-v_* - x_k),$$

$$x_{k+1} \leftarrow \tilde{x}_{k+1} - \log\left(\sum_{i=1}^{n} e^{-\tilde{x}_{k+1}^i}\right).$$

(ii) Wasserstein GF: We approximate numerically the gradient and the laplacian of the log-density:

$$
\begin{aligned}
\forall i \in [n], \quad & (\nabla x_k)^i \leftarrow (x_k^{i+1} - x_k^{i-1})/(2h), \\
\forall i \in [n], \quad & (\Delta x_k)^i \leftarrow (x_k^{i+1} + x_k^{i-1} - 2x_k^i)/h^2, \\
& x_{k+1} \leftarrow x_k + \epsilon(\Delta v_* + \Delta x_k + (\nabla v_* + \nabla x_k)\nabla x_k).
\end{aligned}
\tag{41}
$$

We use periodic boundary conditions, so that the first discretization point is adjacent to the last one for the purposes of computing derivatives.

(iii) Wasserstein-Fisher-Rao GF: We combine the two previous updates. Letting $\nabla x_k$ and $\Delta x_k$ be as in Eq. (41), we have

$$\tilde{x}_{k+1} \leftarrow x_k + \epsilon(-v_* - x_k + \Delta v_* + \Delta x_k + (\nabla v_* + \nabla x_k)\nabla x_k),$$

$$x_{k+1} \leftarrow \tilde{x}_{k+1} - \log\left(\sum_{i=1}^{n} e^{-\tilde{x}_{k+1}^i}\right).$$

We used stepsizes $\epsilon = 2.5 \times 10^{-6}$ and $\epsilon = 1 \times 10^{-6}$ for the experiments on target distributions (1) and (2), respectively. The slopes in Table 1 are obtain by taking $0 < t_1 < t_2$ and computing

$$\frac{\log(\mathrm{KL}(\rho_{t_2}\|\pi)) - \log(\mathrm{KL}(\rho_{t_1}\|\pi))}{t_2 - t_1}.$$

We use different values for $t_1$ and $t_2$ for each target distribution; $t_1$ and $t_2$ must be large enough to capture the asymptotic slope of the curve, but not too large to avoid numerical errors. For all the curves corresponding to target $\pi_1$, we take $t_1 = 7.0$ and $t_2 = 7.5$. For target $\pi_2$, we take: for FR, $t_1 = 6.875$ and $t_2 = 7.0$; for WFR, $t_1 = 1.875$ and $t_2 = 2.0$; for W, $t_1 = 2.75$ and $t_2 = 2.875$.

