# OpenReview forum: "An Explicit Expansion of the Kullback-Leibler Divergence along its Fisher-Rao Gradient Flow"
_TMLR — Accepted by TMLR_

### Review · Reviewer_5Xmo · 2023-03-09

**Summary Of Contributions:**

The paper studies the convergence rate of gradient flow for the KL divergence along the Fisher-Rao dynamics towards its equilibrium at a predetermined target measure.
The main result is that, under very mild assumptions on the target measure and the initial point of the dynamics, the KL divergence decays exponentially fast. Remarkably, the exponent does not depend on either measure, initial, or target. This result improves upon previous works by relaxing the necessary assumptions and by a slight improvement to the convergence rate.
At the technical level, the authors establish an asymptotic expansion of the KL divergence along the dynamics, in terms of the cumulants of the score function.

**Audience:**

Yes

**Broader Impact Concerns:**

No concerns

**Claims And Evidence:**

Yes

**Requested Changes:**

Before outlining my requested changes, let me note that I enjoyed reading the paper and found the result interesting. Since I found the claims made in the submission to be supported by accurate by convincing proofs and since, as indicated above, I believe some individuals in TMLR's audience will be interested in knowing the findings of this paper, I am inclined to recommend acceptance, subject to addressing the issues I raise below.

- Currently, Theorem 1 gives the impression that the error term in the expansion of the KL divergence is bounded by Ce^{-3t}, where C is a *universal* constant. This is also the way the big O notation is presented in the notations paragraph.
As far as I could see from the proof of Theorem 1, this is not the case. C is a constant that depends on the cumulants and on the parameter \alpha, introduced in Assumption A2. In particular, C depends on both the initial and terminal points of the dynamics. I think that this point should be emphasized and discussed.

- In line with my above comment, the discussion about Assumption B is a bit awkward. The existence of the constant M is equivalent to Assumption A2 (when \alpha = 1). The point is that the convergence rate in the cited paper depends explicitly on M. In the present paper, the convergence rate will also depend on M (and \alpha), but only implicitly through the cumulants.

- I found the introduction to be a bit confusing. A lot of emphasis is given to the Wasserstein/Langevin dynamics flow and the Fisher-Rao-Wasserstein flow when the results only deal with the Fisher-Rao flow. While I think that the addition of background and relevant results is great and well-written, it may perhaps be better to separate it from the main focus of the paper. I propose to add a new section about the 'bigger picture' in which the relevant material and corollaries can be presented. (This comment is a matter of personal taste and I leave it to the authors' discretion)

- Line below Equation 11. The superscript W is missing from some of the rhos.

- Statement of theorem 1. I'm not sure what the sentence for "...for t large enough..." is trying to convey. Isn't the expansion valid for any t? The big O notation is supposed to take care if smaller t's. Also I'm not sure why at the end of the proof of Theorem 1, t needs to be taken to infinity to complete the proof. What is needed to show, and what the authors do show, is that the coefficients in the expansion do not grow too fast.

- Equation 32. Is the equation missing the normalizing constants?

**Strengths And Weaknesses:**

Strengths:
- The proof is short, succinct, and well-written.
- Assumption A2 allows the consideration of many new cases, which were unreachable with previous results.

Weaknesses:
- First and foremost, the result is not algorithmic (and probably cannot be made, in full generality). Thus, while this is a fine paper in Wasserstein metric geometry, it is tangentially related to the main topics of TMLR. Still, there is a lot of interest in the Wasserstein gradient flows in the ML community, so I would not hold this point against the paper.
- The improvement in the convergence rate is somewhat incremental compared to previous results. More on this in the next section.

---

> ### Author Response · Authors · 2023-04-03
>
>
> Thank you for your positive feedback!
>
> “At the technical level, the authors establish an asymptotic expansion of the KL divergence along the dynamics, in terms of the cumulants of the score function.” We want to briefly clarify that it is not characterized by score function, but the log-ratio of the initial and final target densities.
>
> “Theorem 1 gives the impression that the error term in the expansion of the KL divergence is bounded by $Ce^{-3t}$. [...]” We clarified this by adding this sentence to Theorem 1: “The remainder terms $O(e^{-3t})$ and $O_q(e^{-3t})$ depend on the initialization $\rho_0$ and on the target $\pi$.” In the paragraph prior to the theorem, we also added a pointer to the equations in the proof that contain the power series expansions for the KL and Renyi divergences.
>
> “[...] The existence of the constant M is equivalent to Assumption A2 (when \alpha = 1). The point is that the convergence rate in the cited paper depends explicitly on M. In the present paper, the convergence rate will also depend on M (and \alpha), but only implicitly through the cumulants.” The reviewer is right in pointing out that the convergence rate stated by Lu et al., 2022 makes use of M in their upper-bound. We rephrased the sentence after the result to reflect this and we removed the part that could confuse the reader.
>
> “I found the introduction to be a bit confusing. [...] I propose to add a new section about the 'bigger picture' in which the relevant material and corollaries can be presented. (This comment is a matter of personal taste and I leave it to the authors' discretion)” The reason that the paper puts emphasis the Wasserstein/Langevin and Wasserstein-Fisher-Rao gradient flows is that the closest prior works approach the study of Fisher-Rao gradient flows from the perspective of sampling, and are more concerned with Wasserstein-Fisher-Rao than with standalone Fisher-Rao, which is cursed by dimension and hence not an implementable algorithm in high dimensions. However, Fisher-Rao is much more amenable to analysis than Wasserstein-Fisher-Rao. In other words, a major motivation to understand the convergence of Fisher-Rao gradient flows is to be able to translate results into WFR gradient flows. We thank the reviewer for their suggestion, but we will stick to the current organization.
>
> “Line below Equation 11. The superscript W is missing from some of the rhos.” We corrected this.
>
> “Statement of theorem 1. I'm not sure what the sentence for "...for t large enough..." is trying to convey. Isn't the expansion valid for any t? [...] Also I'm not sure why at the end of the proof of Theorem 1, t needs to be taken to infinity to complete the proof. [...]” We removed the part "...for t large enough..." from the sentence as we agree that it is redundant. We also changed the last sentence of the proof of Theorem 1 to avoid the confusion.
>
> “Equation 32. Is the equation missing the normalizing constants?” There was a small mistake in this proof that has been corrected. The proof is now correct.

---

### Review · Reviewer_ekbs · 2023-03-13

**Summary Of Contributions:**

The authors drew a very interesting connection between simulated annealing and Wasserstein--Fisher--Rao gradient flow. Using this connect, the authors were able to compute an explicit expansion of the Fisher--Rao gradient flow decay of KL and Renyi divergences. The authors also provided convincing numerical evidence supporting their theoretical results.

**Audience:**

Yes

**Claims And Evidence:**

Yes

**Requested Changes:**

I have several minor questions for clarity.
1. The authors briefly commented on Wasserstein and Fisher--Rao geometry being essentially orthogonal. Can I interpret the WFR Riemannian metric as a sum of Wasserstein and FR separately?
2. If we can take the above interpretation, does the connection to simulated annealing still hold if we vary the coefficients in the linear combination of the two Riemannian metrics?
3. Can we compute a higher order term in the $O(e^{-3t})$ expansion, so that we get a type of Taylor remainder term?
4. In your numerical simulations, can you describe how the gradient flow evolution and KL were calculated? In particular, you chose the support $[-\pi,pi)$, but is the boundary condition period, i.e. making it a torus?

**Strengths And Weaknesses:**

Strengths

1. As far as I know, the connection to simulated annealing is novel, and honestly somewhat surprising. I believe this observation alone is significant.
2. Using the cumulant-generating function to compute a power series expansion is also a new technique based on my knowledge, at least it is not well known and therefore welcomed. The authors were able to use this expansion to show the first order term does indeed decay at the desired rate $e^{-2t}$, which is much cleaner than choosing $\delta,t_*$ in previous works.

Weaknesses

Only several minor clarifications and questions in the next section.

---

> ### Author Response · Authors · 2023-04-03
>
> Thank you for the positive comments!
>
> “The authors briefly commented on Wasserstein and Fisher--Rao geometry being essentially orthogonal.” The two geometries are not really orthogonal, it’s merely a heuristic to understand their behavior. See Section 3 of the citation (Gallouët & Monsaingeon, 2017) for more precise statements; in particular, the distances are infinitesimally orthogonal. In our experimental section, Table 1 shows that the slopes for WFR are very close to the sum of the slopes of Wasserstein and Fisher-Rao, so it would be nice to formalize that in future works if possible, although there doesn’t seem to be a clear avenue towards that goal.
>
> “does the connection to simulated annealing still hold if we vary the coefficients in the linear combination of the two Riemannian metrics?” The connection to simulated annealing only holds for the Fisher--Rao geometry. As we just pointed out, the two geometries are not really orthogonal for WFR. Hence, this approach would break down.
>
> “Can we compute a higher order term in the $O(e^{−3t})$ expansion, so that we get a type of Taylor remainder term?” In the paragraph prior to the theorem, we added a pointer to the equations in the proof that contain the power series expansions for the KL and Renyi divergences. Looking at those equations one can derive expansions to any order. A Taylor expansion with remainder is possible to any order $k$ (because the function is analytic), and it would be interesting to follow this approach if one has a uniform bound on the $(k+1)$-th order cumulants along the linear interpolation between initialization and target, as we could write an explicit bound. However, there is not a clear way to obtain such a bound.
> The full details of our numerical experiments are outlined in Appendix B, and were already there in our submission. We enforce boundary conditions by making it a torus. You can do this by ensuring the first and last components of your vector are adjacent when simulating the transport term for Wasserstein or WFR.

---

> > ### Comment · Reviewer_ekbs · 2023-04-03
> > **Response**
> >
> > Thank you for clarifying my questions. I'm happy with the current state of the paper and will recommend accept.

---

### Review · Reviewer_sQ4p · 2023-03-22

**Summary Of Contributions:**

In this paper, the authors improve the results of the convergence of the Wasserstein Fisher-Rao gradient flow that was recently established by Lu et al. (2019, 2022). The key technique in this paper is the use of an infinite power series expansion related to the cumulant-generating function of random variables.


**Audience:**

Yes

**Claims And Evidence:**

Yes

**Requested Changes:**

In the introduction of the convergence rates for Langevin Monte Carlo sampling algorithms, you might need to elaborate a little bit more on the differences among these works. In particular, it is worth discussing other representing papers such as [1] that directly derive the convergence of discrete-time LMC and [2] that shows the convergence of the stochastic version of LMC with an arbitrary choice of mini-batch size.
[1] Global convergence of Langevin dynamics-based algorithms for nonconvex optimization. Advances in Neural Information Processing Systems, 2018.
[2] Faster convergence of stochastic gradient Langevin dynamics for non-log-concave sampling. In Uncertainty in Artificial Intelligence, 2021.

The authors claim that the results presented in this paper are tight and require fewer assumptions than previous work. It would be helpful if the authors could provide more detail on this point, including the lower bound of the problem and the relaxation of assumptions from previous work. In addition, it would be valuable to discuss how the assumptions are essential to their paper.

In Section 2.4, it is unclear whether $\kappa_n$ represents the n-th moment of Y.

There is a lack of comprehensive comparison between the results presented in this paper (Theorem 1) and those of existing results, particularly Lu et al. (2022). It is noted that the results in Lu et al. (2022) hold for any $\delta\in(0,½]$. Furthermore, if a value is arbitrarily chosen for $\delta$, the result in their paper is also in the same order as that of Theorem 1. Moreover, the term $t^*$ would also be absorbed by the constant $C_1$. Hence, it is suggested that the authors provide a more detailed comparison between their results and those of existing work.

Regarding the proof, it is mentioned that even though the decomposition of the KL divergence only involves terms like $e^{-2t}$ and $e^{-3t}$, the $O(e^{-3t})$ term actually hides problem-dependent constants related to the infinite series expansion. Therefore, it is not directly comparable with Lu et al. (2022).

Minor questions:

* In the abstract: **$\pi$ is that independent of** -> **$\pi$ that is independent of**
* Around Eq (3): **satisfies a LSI** -> **satisfies an LSI**
* Below Eq (3): **when $V^∗$ $\alpha$-strongly convex** -> **when $V^∗$ ``is`` $\alpha$-strongly convex**


**Strengths And Weaknesses:**

The paper is written in a concise way and the proof seems straightforward. The problem studied here is also of great importance. However, clarity over previous work is lacking. In particular, the comparison of the results obtained in this paper with the immediately relevant work on this topic seems not explicit, which makes it hard to judge the contribution of the current paper.

---

> ### Author Response · Authors · 2023-04-03
>
> Thank you for the positive comments.
>
> We want to clarify that this is not a paper that proves rates for Wasserstein-Fisher-Rao, but provides an exact asymptotic expression for the Kullback-Leibler divergence along its Fisher-Rao gradient flow. Similarly, the main contribution of the papers (Lu et al., 2019 & 2022) is their study of the Fisher-Rao gradient flows, not WFR.
>
> “[...] it is worth discussing other representing papers such as [...]” As this is not really a paper about Langevin Monte Carlo (especially not about stochastic mini-batch convergence of LMC), we found that these references are out of scope for this work, though we will keep these in mind in case one of us writes a paper on sampling.
>
> “[...] The authors claim that the results presented in this paper are tight and require fewer assumptions than previous work. It would be helpful if the authors could provide more detail on this point [...]” At the beginning of Section 3, we state explicitly that our results assume A1 and A2, while (Lu et al., 2019 & 2022) work under a subcase of A2 which is much more restrictive.
>
> “In Section 2.4, it is unclear whether $\kappa_n$ represents the $n$-th moment of $Y$.” It is explicitly stated that $\kappa_n$ is the $n$-th cumulant of $Y$ in Section 2.4.
>
> “There is a lack of comprehensive comparison between the results presented in this paper (Theorem 1) and those of existing results [...]” We added the following paragraph after Theorem 1. “Note that the result Eq. (4) by \cite{lu2019accelerating} implies an asymptotic rate very close to $e^{-2t}$, but there are significant differences between both: beyond the fact that our result holds under much weaker assumptions, we characterize exactly the asymptotic decay of $\text{KL}(\rho_t\|\pi)$, while they only provide an upper-bound that becomes less tight as $\delta$ goes to zero.”
>
> Thank you for pointing out the typos; these will be fixed in the final version.

---

> > ### Comment · Reviewer_sQ4p · 2023-04-07
> > **Thank you for the response**
> >
> > Thank you for addressing my previous concerns. However, I still have some minor points that need to be discussed. Firstly, the authors did not explain why they claim the result to be tight, and I suggest they revise the manuscript to clarify this point. I appreciate the added discussion after Theorem 1, and I agree that the result in this paper is obtained in a weaker condition than previous works. However, the authors may have overclaimed the other two contributions. Since the result uses $O(e^{-3t})$, it cannot be claimed to be an exact decay because the KL is only upper bounded by a quantity up to a constant factor. Regarding Lu et al. (2019), it seems that $\delta$ could be fixed arbitrarily, and the comment mentioned in the paper may not be an issue. Additionally, in Lu et al. (2022), the results do not depend on $\delta$, and I suggest the authors compare their work with this paper as well. Overall, I appreciate the authors' efforts and encourage them to address these minor points in the revised manuscript.

---

> > > ### Author Response · Authors · 2023-04-07
> > >
> > > 1. Our result is “tight” insofar as we proved an exact expansion of the KL divergence under the FR flow that is considered in Lu et al. (2019,2022). The first order term dominates, yielding the tight exponent, and the next-order terms are smaller for large times. This motivates our use of big-O notation, which we clarified in the main text as per the request of the other reviewers.
> > >
> > > 2. Yes, $\delta$ can be fixed arbitrarily in their paper, yielding an upper-bound expansion that is not tight for large times.
> > >
> > > 3. In the follow-up work of Lu et al. (2022), the authors abandon an approach that yields tight exponents in the large-time regime, but instead have an upper bound expansion for all time with no warm-start. Note that in the limit, this expansion does not yield the correct rate of convergence, but once the KL is sufficiently small, they can use their previous results with the warm-start.
> > >
> > > We hope this clarifies your concerns.

---

### Decision · Action_Editors · 2023-05-04

**Recommendation:** Accept as is

**Comment:**

Three expert reviewers reviewed the paper and they all had positive evaluation, which AC agrees with. This is a nice theoretical paper with great presentation style which makes it accessible to a wider ML audience. AC recommends accepting this paper for publication.

**Audience:**

The theoretical results in this paper are insightful and of interest to TMLR community.

**Claims And Evidence:**

The paper draws connections between simulated annealing and Wasserstein--Fisher--Rao gradient flow and computes a Fisher--Rao gradient flow decay in KL and Renyi divergences. The presentation is clear and proofs seem to be written with care. The paper also includes numerical experiments demonstrating the theoretical results.

Three expert reviewers reviewed the paper and they all had positive evaluation, which AC agrees with. AC recommends accepting this paper for publication.